# Spatial Galloping Behavior of Iced Conductors under Multimodal Coupling

**DOI:** 10.3390/s24030784

**Published:** 2024-01-25

**Authors:** Fujiang Cui, Kaihong Zheng, Peng Liu, Han Wang

**Affiliations:** 1Beijing Aircraft Technology Research Institute, Commercial Aircraft Corporation of China, Ltd., Beijing 102211, China; cuifujiang@tyut.edu.cn; 2College of Aeronautics and Astronautics, Taiyuan University of Technology, Taiyuan 030024, China; liupeng01@tyut.edu.cn (P.L.); wanghan@tyut.edu.cn (H.W.)

**Keywords:** iced conductors, multimodal coupling, galloping behavior, nonlinear vibration

## Abstract

In this study, the coupled ordinary differential equations for the galloping of the first two modes in iced bundled conductors, including in-plane, out-of-plane, and torsional directions, are derived. Furthermore, through numerical analysis, the critical conditions of this modal galloping are determined in the range of wind speed–sag parameters, and the galloping patterns and variation laws in different parameter spaces are analyzed. The parameter space is then divided into five regions according to the different galloping modes. Under the multimodal coupling mechanism of galloping, the impact of single and two kinds of coupled mode galloping on the spatial nonlinear behavior is explored. The results reveal that the system exhibits an elliptical orbit motion during single mode galloping, while an “8” motion pattern emerges during coupled mode galloping. Moreover, two patterns of “8” motion are displayed under different parameter spaces. This research provides a theoretical foundation for the design of transmission lines.

## 1. Introduction

The cross-sectional shape of iced conductors changes from circular to noncircular under complex climatic conditions such as ice storm, snow, and freezing rain. Then, under the excitation of a certain wind speed, the unstable aerodynamic nonlinear load in different directions, including in-plane, out-of-plane, and torsional directions, induces galloping of the iced conductor. It is a self-excited vibration phenomenon characterized by a low frequency (approximately 0.08–3 Hz) and a large amplitude (reaching 5–300 times the diameter of the conductor) [1]. The galloping lasts for several hours and can cause serious damage to the power conductors, including tower damage, conductor fracture, insulator damage, wear of fittings and components, and phase flashover, which pose a significant threat to the safety of power conductors.

Until now, several researchers have explored the excitation mechanism of galloping, mainly focusing on the aerodynamic coefficients of iced conductors, line structure parameters, etc. Further, the galloping phenomenon has been examined from different perspectives. The three typical galloping mechanisms include vertical galloping [2], torsional galloping [3], and inertially coupled galloping [4]. In recent years, researchers from the China Electric Power Research Institute have investigated numerous domestic and foreign galloping accidents and classified galloping as a dynamic instability phenomenon. It has been reported that only unstable vibration can lead to large galloping. Accordingly, the dynamic stability mechanism of galloping has been proposed to analyze different types of galloping [5,6]. Based on these mechanisms, galloping has been explored through theoretical analysis.

Due to the geometric nonlinearity of iced conductor structures and the aerodynamic nonlinearity caused by aerodynamic loads, nonlinear coupling between multiple directions/modes of galloping is very likely to occur. Jones [7] developed a dynamic model for the coupling of in-plane and out-of-plane galloping and found that there is an inherent coupling between the equations describing in-plane and out-of-plane motions. This model demonstrated that out-of-plane motion has a significant impact on the stability of galloping. Blevins and Iwan [8] developed a dynamic model considering the coupling of torsional and in-plane motions, and they comprehensively analyzed galloping under both resonant and nonresonant conditions. This study was the first to prove that torsional motion can enhance galloping in the system. Liu et al. [9] used Hamilton’s principle to establish three types of dynamic models for coupled galloping with in-plane, in-plane+torsional, and in-plane+out-of-plane+torsional motions. By analyzing the in-plane amplitude and torsional angle of galloping under various influencing factors, such as wind speed, air density, span length, damping ratio, and initial tension, they found that the coupled dynamic model of in-plane+out-of-plane+torsional motion was more accurate in evaluating the galloping characteristics. Chen and Wu [10] then clarified the generation mechanism of various galloping vibrations with different coupled motions. Matsumiya et al. [11] indicated the coupling effects on in-plane oscillation by considering the energy balance of the in-plane motion with the defined amplitudes and phase differences of the out-of-plane and torsional motions. In addition, the finite element method is widely used to simulate the galloping of iced conductors. Diana et al. [12] presented a finite-element model of quad-bundled conductors, predicting the onset speed of galloping instability and the maximum oscillation amplitudes through time-domain simulations and the proposed energy approach. The complex structural part of the system can also be reproduced, including iced eight-bundled conductors [13,14] and a tower line system [15]. Complex wind field conditions can also be simulated, such as unsteady and stochastic wind fields [16].

Most studies have focused on the first-order modes in different directions. However, Liu and Huo [17] included higher-order modes and established the coupled motion equations for the first four in-plane modes and the first torsional mode to describe the nonlinear interactions between the in-plane and torsional vibration, considering geometrical and aerodynamic nonlinearities. They found that the galloping of higher-order modes can excite the galloping of lower-order modes, and the energy transfer phenomenon between symmetric and anti-symmetric modes was also analyzed. Huo et al. [18] established the coupled motion equations for the first three in-plane modes and the first three torsional modes, and the results revealed that the torsional modes contributed to the in-plane galloping behavior. With the increase in wind speed, the lower-order in-plane modes were gradually replaced by higher-order in-plane modes. Luongo and Nayfeh [19,20] also pointed out that higher-order modes exist, and these modes can excite the galloping of lower-order modes during the vibration of suspended cables. For such suspended structures, due to the additional tension caused by motion-induced deformation, the natural frequency of the in-plane symmetric mode exceeds that of the anti-symmetric mode, resulting in a frequency crossover phenomenon [21,22,23]. The various correspondence relationships between frequencies provide multiple possibilities for the interaction between modes. Accordingly, in some previous studies, the first two modes of in-plane, out-of-plane, and torsional motion were considered, and the nonlinear coupling between in-plane+out-of-plane+torsional modes and high-order+low-order modes was investigated from the perspective of energy transfer, clarifying the multimode coupling galloping mechanism of iced conductors. Under the coupling effect of multiple modes, the galloping behavior of iced conductors is bound to become complex and diverse. Therefore, exploring the galloping behavior under the coupling of multiple modes is of immense theoretical significance for clearly understanding the galloping of iced conductors and formulating effective anti-galloping measures.

To this end, in this study, the multimodal coupling mechanism of galloping in iced conductors is utilized to investigate the spatial nonlinear behavior of galloping in iced bundled conductors. The rest of this paper is organized as follows. In Section 2, for the common iced bundled crescent-shaped conductors, considering geometric and aerodynamic nonlinearity, the coupled vibration ordinary differential equations for the first two modes in the in-plane, out-of-plane, and torsional directions are derived. In Section 3, through numerical analysis, the critical conditions for in-plane galloping are obtained in the wind speed–span parameter space. Combined with the multimodal coupling galloping mechanism, the influence of single-mode galloping and coupled-mode galloping on spatial galloping behavior is analyzed for different galloping patterns within the parameter domain. Finally, this study is concluded in Section 4.

## 2. Establishment of Dynamical Equations

Compared to a single conductor, the bundled conductors are affected by spacers, and the twisting stiffness of its sub-conductors is much higher than that of a single conductor with the same cross-section. This leads to a more irregular cross-sectional shape of the iced bundled conductors, and the aerodynamic load acting on them is more complex and prone to causing vibrations. Therefore, the common crescent-shaped iced quad-bundled conductor is taken as the research object here.

The iced quad-bundled conductors are simplified as a single conductor for analysis assuming both ends to be fixed. A detailed description of the galloping model is reported in [24]; the main details are hereafter presented. In actual engineering, the sag-to-span ratio of transmission lines is generally small, within the range from 0 to 0.1. The conductor has a slender and flexible body structure, so the influence of bending stiffness can be neglected. In addition, it is assumed that the ice is uniformly distributed along the surface of the conductor with a length of *l*, and the incoming wind blows perpendicular to the plane where the conductor is located with a speed *U*. A schematic of the spatial model is shown in Figure 1a. The initial configuration of the iced conductor under its own weight is represented by $\Gamma_{0}$, while the dynamic configuration of the iced conductor under aerodynamic loads is represented by Γ. $u\left(x,t\right)$, $v\left(x,t\right)$, $w\left(x,t\right)$, and $\theta \left(x,t\right)$ represent the axial (*x*-axis), in-plane (*y*-axis), out-of-plane (*z*-axis), and torsional (*yOz* plane) displacement, respectively, of a point on the iced conductor at time *t* with respect to the coordinate origin *O*. The differential element d*x* is studied, and a schematic of its motion is shown in Figure 1b, where *A*0*B*0 and *A*2*B*2 represent the differential element before and after structural deformation, respectively. The catenary equation of the structure at time t=0 is expressed as follows:(1)$$y_{0}=\frac{-2H}{mg}\sinh\left(\frac{mgx}{2H}\right)\sinh\left(\frac{mg\left(l-x\right)}{2H}\right)$$
where *H* is the initial horizontal tension of the iced conductor, *m* is the mass per unit length, and *g* is the gravitational acceleration.

The aerodynamic force analysis diagram of the iced conductor cross-section is shown in Figure 1c, where *G* is the centroid, $e_{r}$ is the eccentricity, $\theta_{0}$ is the initial wind angle of attack, *I* is the moment of inertia about point *O*, and the overdot denotes the derivative with respect to time. $F_{L}$, $F_{D}$, and M are the aerodynamic lift force, drag force, and torsional moment, respectively. $\alpha_{0}$ is the angle caused by vertical velocity, α is the angle of attack, U is the mean wind speed, $U_{r}$ is the relative wind speed, and D is the bare conductor diameter.

Considering the geometric and aerodynamic nonlinearity and simplifying the axial motion to the in-plane and out-of-plane directions, the coupled vibration ordinary differential equations of the first two modes, including in-plane, out-of-plane, and torsional modes, can be derived using Hamilton’s principle and the Galerkin space-discretization method [24], i.e.,
(2)$$\begin{array}{l} {\ddot{q}}_{vk}+g_{vk,1}{\ddot{q}}_{\theta k}+2\xi_{vk}\omega_{g,vk}{\dot{q}}_{vk}+\omega_{g,vk}^{2}q_{vk}=g_{vk,2}q_{w2}^{2}+g_{vk,3}q_{w1}q_{w2}+g_{vk,4}q_{w1}^{2}\\ \;\;\;\;\;+g_{vk,5}q_{v2}^{2}+g_{vk,6}q_{v1}q_{v2}+g_{vk,7}q_{v1}^{2}+g_{vk,8}q_{v2}q_{w2}^{2}+g_{vk,9}q_{v2}q_{w1}q_{w2}\\ \;\;\;\;\;+g_{vk,10}q_{v2}q_{w1}^{2}+g_{vk,11}q_{v2}^{3}+g_{vk,12}q_{v1}q_{w2}^{2}+g_{vk,13}q_{v1}q_{w1}q_{w2}+g_{vk,14}q_{v1}q_{w1}^{2}\\ \;\;\;\;\;+g_{vk,15}q_{v1}q_{v2}^{2}+g_{vk,16}q_{v1}^{2}q_{v2}+g_{vk,17}q_{v1}^{3}+g_{vk,18}{\dot{q}}_{\theta 1}^{2}+g_{vk,19}{\dot{q}}_{\theta 2}^{2}+g_{vk,20}{\dot{q}}_{\theta 1}{\dot{q}}_{\theta 2}\\ \;\;\;\;\;+\int_{0}^{l}\left[\varphi_{vk}(x)F_{vk}\left({\dot{q}}_{v1},{\dot{q}}_{v2},{\dot{q}}_{w1},{\dot{q}}_{w2},q_{\theta 1},q_{\theta 2},{\dot{q}}_{\theta 1},{\dot{q}}_{\theta 2}\right)\right]dx,\;k=1,2\\ {\ddot{q}}_{wk}+g_{wk,1}{\ddot{q}}_{\theta k}+2\xi_{wk}\omega_{g,wk}{\dot{q}}_{wk}+\omega_{g,wk}^{2}q_{wk}=g_{wk,2}q_{v2}q_{w2}+g_{wk,3}q_{v2}q_{w1}\\ \;\;\;\;\;+g_{wk,4}q_{v1}q_{w2}+g_{wk,5}q_{v1}q_{w1}+g_{wk,6}q_{w2}^{3}+g_{wk,7}q_{w1}q_{w2}^{2}+g_{wk,8}q_{w1}^{2}q_{w2}\\ \;\;\;\;\;+g_{wk,9}q_{w1}^{3}+g_{wk,10}q_{v2}^{2}q_{w2}+g_{wk,11}q_{v2}^{2}q_{w1}+g_{wk,12}q_{v1}q_{v2}q_{w2}+g_{wk,13}q_{v1}q_{v2}q_{w1}\\ \;\;\;\;\;+g_{wk,14}q_{v1}^{2}q_{w2}+g_{wk,15}q_{v1}^{2}q_{w1}+g_{wk,16}{\dot{q}}_{\theta 1}^{2}+g_{wk,17}{\dot{q}}_{\theta 2}^{2}+g_{wk,18}{\dot{q}}_{\theta 1}{\dot{q}}_{\theta 2}\\ \;\;\;\;\;+\int_{0}^{l}\left[\varphi_{wk}(x)F_{wk}\left({\dot{q}}_{v1},{\dot{q}}_{v2},{\dot{q}}_{w1},{\dot{q}}_{w2},q_{\theta 1},q_{\theta 2},{\dot{q}}_{\theta 1},{\dot{q}}_{\theta 2}\right)\right]dx,\;k=1,2\\ {\ddot{q}}_{\theta k}+g_{\theta k,1}{\ddot{q}}_{vk}+g_{\theta k,2}{\ddot{q}}_{wk}+2\xi_{\theta k}\omega_{g,\theta k}{\dot{q}}_{\theta k}+\omega_{g,\theta k}^{2}q_{\theta k}=+g_{\theta k,3}{\dot{q}}_{v1}{\dot{q}}_{\theta 1}+g_{\theta k,4}{\dot{q}}_{v1}{\dot{q}}_{\theta 2}\\ \;\;\;\;\;+g_{\theta k,5}{\dot{q}}_{v2}{\dot{q}}_{\theta 1}+g_{\theta k,6}{\dot{q}}_{v2}{\dot{q}}_{\theta 2}+g_{\theta k,7}{\dot{q}}_{w1}{\dot{q}}_{\theta 1}\;+g_{\theta k,8}{\dot{q}}_{w1}{\dot{q}}_{\theta 2}+g_{\theta k,9}{\dot{q}}_{w2}{\dot{q}}_{\theta 1}+g_{\theta k,10}{\dot{q}}_{w2}{\dot{q}}_{\theta 2}\\ \;\;\;\;\;+\int_{0}^{l}\left[\varphi_{\theta k}(x)M_{\theta k}\left({\dot{q}}_{v1},{\dot{q}}_{v2},{\dot{q}}_{w1},{\dot{q}}_{w2},q_{\theta 1},q_{\theta 2},{\dot{q}}_{\theta 1},{\dot{q}}_{\theta 2}\right)\right]dx,\;k=1,2 \end{array}$$
where $q_{vk}\left(t\right)$, $q_{wk}\left(t\right)$, and $q_{\theta k}\left(t\right)$ are the generalized coordinates for in-plane, out-of-plane, and torsional motions, respectively. $\xi_{vk}$, $\xi_{wk}$, $\xi_{\theta k}$; $\omega_{g,vk}$, $\omega_{g,wk}$, $\omega_{g,\theta k}$; and $\varphi_{wk}$, $\varphi_{vk}$, and $\varphi_{\theta k}$ are the damping ratios, natural frequencies, and natural modes corresponding to the different order modal structures for in-plane, out-of-plane, and torsional directions, respectively. $g_{vi,k}(i=1,\ldots ,22)$, $g_{wi,k}(i=1,\ldots ,20)$, and $g_{\theta i,k}\left(i=1,\ldots ,12\right)$ are the integral coefficients. $\int_{0}^{l}\left[\varphi_{wk}(x)F_{wk}\right]dx$, $\int_{0}^{l}\left[\varphi_{vk}(x)F_{vk}\right]dx$, and $\int_{0}^{l}\left[\varphi_{\theta k}(x)M_{\theta k}\right]dx$ are the aerodynamic terms for in-plane, out-of-plane, and torsional directions, respectively [24].

## 3. Numerical Analysis

The multimodal coupling mechanism of galloping is described [24] as follows: the nonlinear interaction among the in-plane, out-of-plane, and torsional modes of the same order leads to synchronized and limited galloping phenomena. The nonlinear coupling between the torsional and in-plane modes is the fundamental reason for the limited galloping characteristics. Regarding the synchronized galloping characteristics, the in-plane, out-of-plane, and torsional modes of the same order are excited simultaneously and tend to stabilize simultaneously, among which the galloping in the in-plane direction plays a dominant role. Regarding the limited galloping characteristics, the galloping is limited to a certain amplitude range. The multimodal coupling effect of galloping directly affects its spatial nonlinear behavior.

### 3.1. Critical Conditions of Galloping

When the span and the unit mass per unit length of the conductor are determined, the relationship between the sag (*d*) and the initial tension in the horizontal direction can be obtained as follows:(3)$$d=\frac{2H}{mg}{\left[\sinh(\frac{mgl}{4H})\right]}^{2}$$

During the actual installation process of power lines, the sag of the conductors must be controlled based on different terrain sections to ensure a safe discharge distance. Therefore, sag is chosen as a bifurcation parameter to examine the critical conditions for galloping. The nonlinear dynamical equations (Equation (2)) are used to numerically analyze the unstable regions of the first two modes in different directions. Further, the critical instability conditions for the in-plane, out-of-plane, and torsional modes are explored in the wind speed (*U*) and sag (*d*) parameter plane. The common crescent-shaped iced quad-bundled conductors 4 × LGJ-400/35 are taken as an example in this study, and the selected equivalent parameters are shown in Table 1 [25].

Taking in-plane motion as an example, the critical conditions and galloping region of the first two modes in the *U-d* parameter plane are shown in Figure 2. The blue and red lines represent the critical conditions of the first- and second-order modal galloping, respectively. According to this analysis, the following conclusions can be drawn:(1)For the first-order modal galloping, as the sag increases, the vibration area first increases and then gradually decreases. When *d* > 3.8 m, the vibration area begins to decrease rapidly until the vibration disappears. However, when *d* > 5.6 m, the first-order modal vibration is again excited because of the nonlinear coupling between the first- and second-order modes in the in-plane direction [24]. Within the range of 5.6 m < *d* < 6.8 m, as the sag increases, the critical wind speeds for the upper and lower limits of the vibration increase rapidly, and the vibration area is narrow but slowly increasing. When *d* > 6.8 m, as the sag continues to increase, the vibration area first gradually increases, reaching a maximum around *d* = 8 m, and then begins to gradually decrease. When *d* > 10 m, the first-order modal vibration is no longer excited.(2)When the sag is low, the second-order modal galloping is not excited if the wind speed is within 20 m/s. When *d* > 3.8 m, the first-order modal galloping area begins to decrease, and the second-order modal galloping starts to be excited. The galloping area rapidly expands with the increase in sag. When *d* > 5 m, the galloping area enters a slow expansion stage. It can also be observed that the area of second-order modal galloping is significantly larger than that of first-order modal galloping, indicating that the second-order modal galloping is more likely to occur.

According to the different vibration patterns, the *U-d* parameter plane is divided into five regions. Region I is the stable region where no galloping occurs. Region II and Region III are both single-mode galloping regions, corresponding to single first-order modal and single second-order modal galloping, respectively. Regions IV and V have the same galloping mode, which arises from the coupling of first- and second-order modes.

### 3.2. Galloping Behavior

Now, the spatial galloping behavior of the five galloping regions is separately studied. The in-plane galloping of the iced conductor is a self-excited vibration, and the vibration frequency is basically the same as the natural frequency. Therefore, combined with the multimodal coupling mechanism of galloping, by comparison with the natural frequency, it can be numerically verified whether various order modes in the in-plane, out-of-plane, and torsional directions exhibit galloping and instability. The natural frequencies of the first two modes of in-plane, out-of-plane, and torsional motion at different inclinations are listed in Table 2.

In Region I, for any given nonzero initial displacement, the vibration amplitudes of the first two modes in all the directions rapidly decay to zero, as shown in Figure 3, and there is no modal vibration in this region. Keeping the sag constant, as the wind speed gradually increases, the vibrations begin to enter Region II, and the system starts to become unstable. When the wind speed increases to 14 m/s, the time history and spectral responses of the first two modes in the in-plane, out-of-plane, and torsional directions are shown in Figure 4. It can be seen that only the first-order mode in each direction is excited, and thus each direction exhibits a single first-order mode vibration (Figure 4a). According to Figure 4b–d, the frequencies of the first-order vibrations in all the directions are 0.402 Hz, which is essentially the same as the first natural frequency in the in-plane direction (see Table 2), indicating synchronous vibration characteristics, and the vibration of the first-order mode in the in-plane direction plays the dominant role.

The galloping behavior within Region II is shown in Figure 5. As only the first-order modes are excited in the in-plane, out-of-plane, and torsional directions, the time history responses at 1/2 span for all the directions are stable periodic signals (Figure 5a). The galloping trajectory has only one peak in both in-plane and out-of-plane directions, exhibiting a standard first-order galloping profile (Figure 5b). When observing from the axial direction, the set of galloping trajectories in the entire span is elliptical (Figure 5c), and the galloping trajectory at 1/2 span is an inclined ellipse, showing repetitive motion on the same elliptical trajectory, as shown in Figure 5d. Therefore, the system vibrates along an inclined elliptical trajectory at all points within this region. Consequently, under the coupling effect of in-plane and out-of-plane motions, the spatial galloping trajectory has an approximately inclined elliptical spherical shape.

Keeping the sag constant and further increasing the wind speed, the galloping behavior gradually enters Region III. When the wind speed increases to 19 m/s, the time history and spectral responses of the first two modal profiles in the in-plane, out-of-plane, and torsional directions are shown in Figure 6. In this case, the first two modal profiles in the torsional direction tend to be in a stable state. However, in the in-plane and out-of-plane directions, only the second-order modal profile is excited, and the corresponding galloping frequency in each direction is 0.552 Hz (Figure 6b,c), which is basically the same as the second-order natural frequency in the in-plane direction (see Table 2).

The galloping behavior in Region III is shown in Figure 7. Since only the second-order modes are excited in both the in-plane and out-of-plane directions, the node is located at 1/2 span, and the time history responses at 1/4 span are stable periodic signals (Figure 7a). In this case, the galloping trajectories in both the in-plane and out-of-plane directions have two peaks, indicating a standard second-order galloping profile, as shown in Figure 7b. Observing from the axial direction, similar to Region II, the set of galloping trajectories for the entire iced conductor has an elliptical profile (Figure 7c). The galloping trajectory at the 1/4 span position is a tilted ellipse, and it moves cyclically on the same elliptical trajectory, as shown in Figure 7d. Therefore, it can be concluded that except for the node, all points on the conductor in Region III are vibrating along a tilted elliptical trajectory. Consequently, under the coupling of in-plane and out-of-plane motion, the spatial galloping trajectory also shows two approximately tilted elliptical spheres with a stationary node between them.

Compared with Region II, in Region III, the galloping amplitude in the out-of-plane direction is larger, resulting in a greater tilt of the motion trajectory. It should be noted that in Region III, the phase difference between in-plane and out-of-plane motion is basically 180° (Figure 7a), which is larger than the difference in Region II. This results in a narrower short axis of the elliptical motion at different span positions, appearing to move along a diagonal line. Meanwhile, it can be seen that since single-mode galloping occurs in all the directions of Region II and III, except for the node, the iced conductor at each point follows its own elliptical trajectory of periodic motion.

Until now, the galloping behavior in different regions with single-mode galloping has been analyzed. Next, the galloping behavior in Region IV and V, which both have coupled-mode galloping, are examined. It can be seen in Figure 2 that Region IV is the transition stage between the in-plane first-order and in-plane second-order modal galloping, with a narrow galloping area. When *d* = 4.6 m and *U* = 5.5 m/s, the time history and spectral response of the first two modes in the in-plane, out-of-plane, and torsional directions are shown in Figure 8. In this case, the first mode with a galloping frequency of 0.429 Hz (Figure 8b) is only excited in the in-plane direction, which is basically the same as the first in-plane natural frequency in Table 1 and Table 2. The second mode is excited in both the in-plane and out-of-plane directions, and their galloping amplitudes are basically the same. Their galloping frequency is also 0.519 Hz (Figure 8c,d), which is basically the same as the second in-plane natural frequency in Table 1. The first two modes in the torsional direction are essentially stable.

The galloping behavior in Region IV is shown in Figure 9. Because the node of the second-order mode is located at 1/2 span, only the in-plane first-order mode component exists at 1/2 span (Figure 9a). At 1/4 span, due to the coupling between the two in-plane modes, the time history response is no longer a stable periodic signal and beats occur. The out-of-plane galloping only involves a single second-order mode and the response remains a stable periodic signal (Figure 9b). The galloping trajectory and its projections in one period of the entire span in different directions are shown in Figure 9c. Due to the coupling effect of the modes, there is no fixed peak and node in the in-plane galloping trajectory, but the overall galloping trajectory exhibits an anti-symmetric profile. In the out-of-plane direction, the galloping trajectory has two peaks and one node, exhibiting a standard second-order galloping profile.

Compared with the single-mode galloping regions, the trajectory in the axial direction is no longer a single ellipse, but rather a superposition of multiple inclined ellipses (Figure 9d). The conductor moves along an elliptical path at 1/2 span, but its galloping trajectory is no longer the same elliptical track, as shown in Figure 9e. However, due to the coupling between the modes, the conductor no longer moves along a single galloping trajectory at 1/4 span, as shown in Figure 9f,g. The conductor first moves in an inclined ellipse along the blue line, then in an inclined “8” profile along the black line, and then exhibits an inclined elliptical motion along the red line. Therefore, it can be inferred that in this case, the galloping trajectory exhibits a mixed motion pattern of ellipse and “8” profile due to the coupling effect of multiple modes. Hence, the spatial trajectory of galloping is anti-symmetrical and has no fixed profile.

As the parameters continue to change, the galloping enters Region V. When *d* = 6.8 m and *U* = 14 m/s, the time history and spectral responses of the first two modes in the in-plane, out-of-plane, and torsional directions are shown in Figure 10. It can be seen that all the modes except the second torsional mode are excited and there are dense sidebands at each galloping frequency. Among them, the galloping frequency of the first in-plane mode is the same as that of the second in-plane mode, both of which are dominated by 0.473 Hz (Figure 10b,c). Compared to Table 2, it is found that this frequency is the average of the first and second intrinsic frequencies of the in-plane mode, which may be due to the nonlinear coupling between the first and second in-plane modes, leading to frequent energy exchange between them. In addition, the dominant frequency of the first-order mode in the out-of-plane and torsional directions is 0.237 Hz (Figure 10d–f)). This frequency is half of the in-plane first-order mode frequency due to the synchronization among the three first-order modes [24]. The dominant frequency of the out-of-plane second-order mode is 0.473 Hz (Figure 10e), caused by the synchronization of the three second-order modes [24].

The galloping behavior of the system in Region V is shown in Figure 11. Although there is a node of the second-order mode at 1/2 span, the time history responses are no longer stable periodic signals at this location due to strong coupling between the in-plane first- and second-order modes, resulting in beat phenomenon (Figure 11a). A similar phenomenon occurs at 1/4 span with the coupling between different order modes (Figure 11b). The galloping trajectory and its projection over a single cycle in different directions are shown in Figure 11c. Due to nonlinear coupling between the modes, there are no fixed peaks and nodes in both in-plane and out-of-plane directions, but the overall trajectory has an anti-symmetric profile.

From the axial direction, the galloping trajectories of the entire iced conductor in Region III are more complex than those in Region IV (Figure 11d). However, in this case, the galloping trajectories of each point on the iced conductor have an approximately horizontal “8” shape. Figure 11e shows the galloping trajectory at 1/2 span, where the conductor continuously moves in an “8” pattern without any transition, and the trajectory does not repeat at each cycle. The same is true for the galloping trajectory at 1/4 span (Figure 11f). In this case, the spatial trajectory is also roughly anti-symmetric but without a fixed profile. Moreover, compared to the other galloping regions, it has a wider spatial range, especially in the out-of-plane direction, which can exacerbate wear and tear of the conductor, leading to strand and line breakage as well as damage to the insulator strings and their connections at both ends of the conductor. Therefore, it is necessary to avoid the routing of transmission lines in such situations.

In summary, from the axial direction, as the wind speed increases, the galloping trajectories gradually change from an elliptical shape to an approximately “8” shape. The results are basically consistent with those obtained by the finite element method [13,15] and experiments [26].

## 4. Conclusions

In this study, a simplified dynamic model was established for analyzing the galloping of iced quad-bundled conductors in the in-plane, out-of-plane, and torsional directions. Using numerical analysis in the wind speed–span parameter space, the critical conditions for the first- and second-order modal galloping were investigated, and the galloping modes and their variations in different parameter space were analyzed. The parameter space was divided into five regions based on different galloping patterns, and the spatial nonlinear behavior of system galloping in different regions was discussed under the existence of the multimodal coupling mechanism. The main results of this study are summarized as follows:

(1)As the wind speed and span increased, the galloping process of the system underwent the following sequential stages: first-order mode, coupling of first- and second-order modes, second-order mode, coupling of first- and second-order modes, and second-order mode. Further, first-order mode galloping was excited twice. Moreover, the area of the second-order mode galloping was significantly larger than that of the first-order mode galloping, making it easier to occur in practical situations.(2)When the system was in a single-modal galloping state in all the directions, it exhibited a stable periodic motion. Under the coupled effect of in-plane and out-of-plane motion, except at the nodes, all the points on the iced conductor moved along a continuous, overlapping, and inclined elliptical orbit. When the system was in a single first-order mode galloping state, the spatial trajectory of the galloping motion was an approximately inclined elliptical sphere. When the system was in a single second-order mode galloping state, the spatial trajectory of galloping was approximately two inclined elliptical spheres with an immobile node in the center.(3)When the system was in a coupled-mode galloping state in certain directions, the spatial trajectory of galloping was basically anti-symmetric but had no fixed profile. During the first excitation stage of the first-order mode, the iced conductor moved along a continuous, inclined, elliptical orbit at the 1/2 span position, while at the other points, there was a mixed motion pattern of inclined elliptical and “8” profiles. During the second excitation stage of the first-order mode, all the points on the iced conductor vibrated along a continuous, approximately horizontal “8” profile.

## Figures and Tables

**Figure 1 sensors-24-00784-f001:**
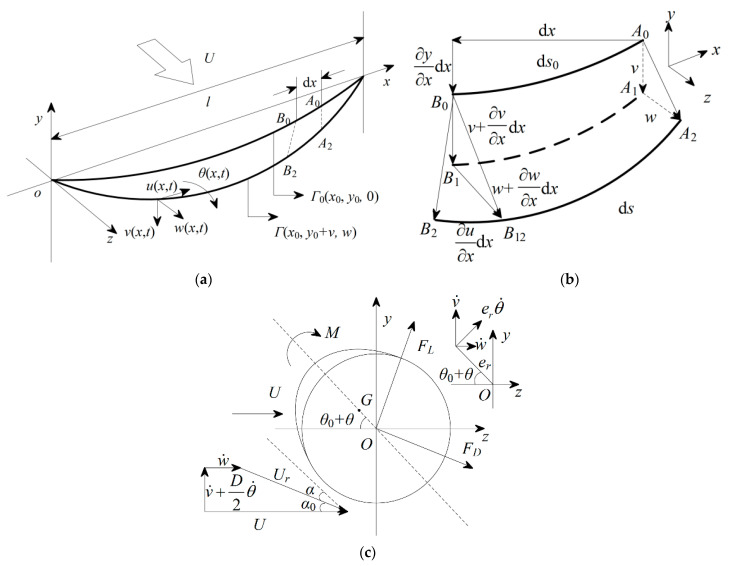
Galloping model of iced conductor. (**a**) Configuration. (**b**) Dynamic displacement of d*x*. (**c**) Aerodynamic forces acting on the chosen section of the iced conductor.

**Figure 2 sensors-24-00784-f002:**
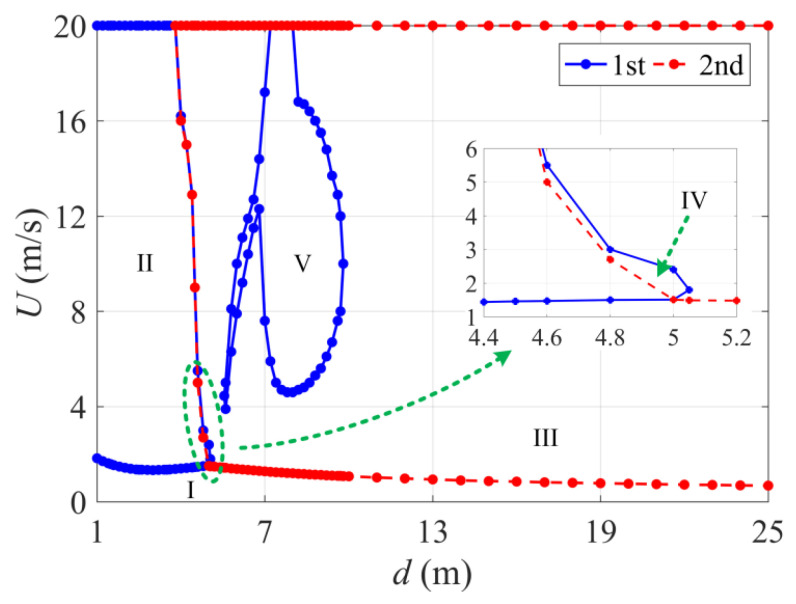
Critical conditions and galloping region division of first- and second-order modal galloping in the *U*-*d* parameter plane.

**Figure 3 sensors-24-00784-f003:**
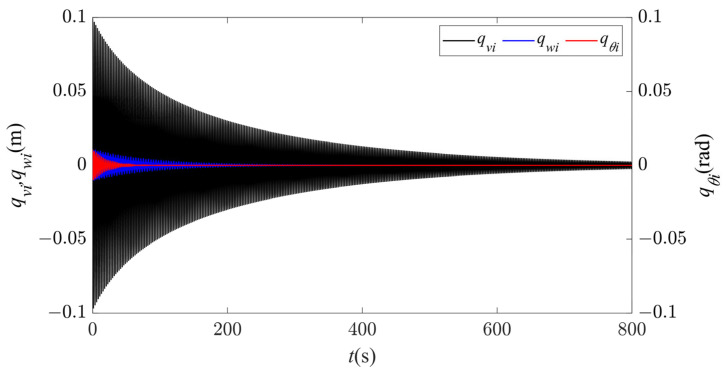
Time history responses of first- and second-order modal galloping along the three directions in Region I (*d* = 4.3 m, *U* = 1 m/s).

**Figure 4 sensors-24-00784-f004:**
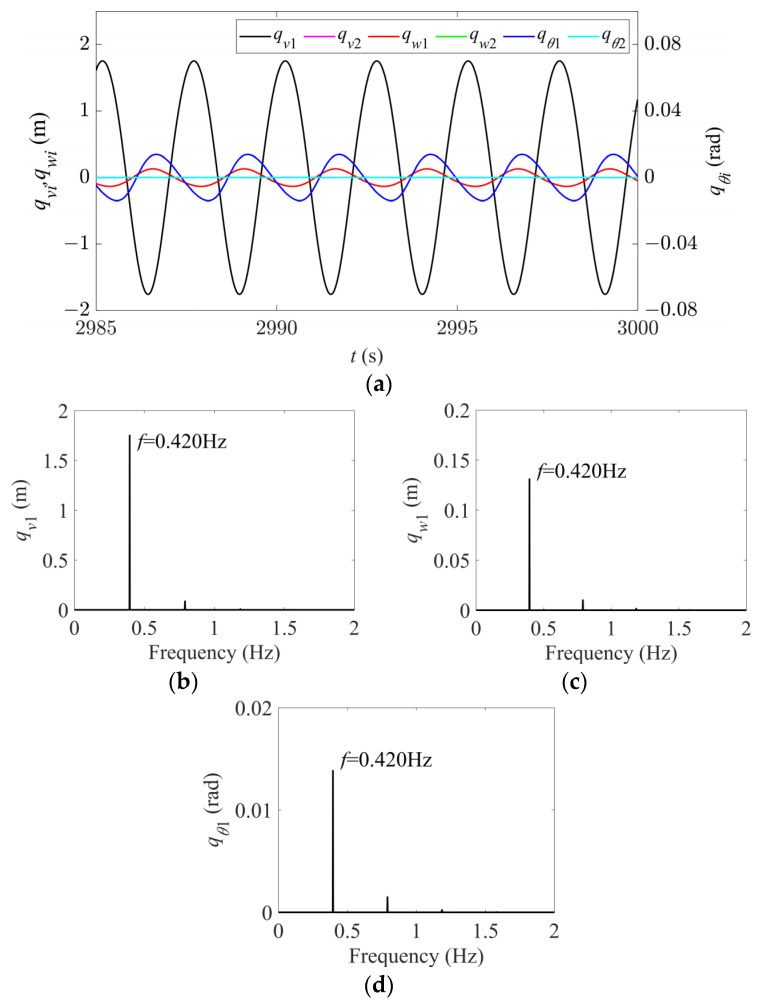
Time history and spectral responses of first- and second-order modal galloping along the three directions in Region II (*d* = 4.3 m, *U* = 14 m/s). (**a**) Time history responses of first- and second-order modal galloping along the three directions. (**b**) Spectral response of first-order modal galloping along the in-plane direction. (**c**) Spectral response of first-order modal galloping along the out-of-plane direction. (**d**) Spectral response of first-order modal galloping along the torsional direction.

**Figure 5 sensors-24-00784-f005:**
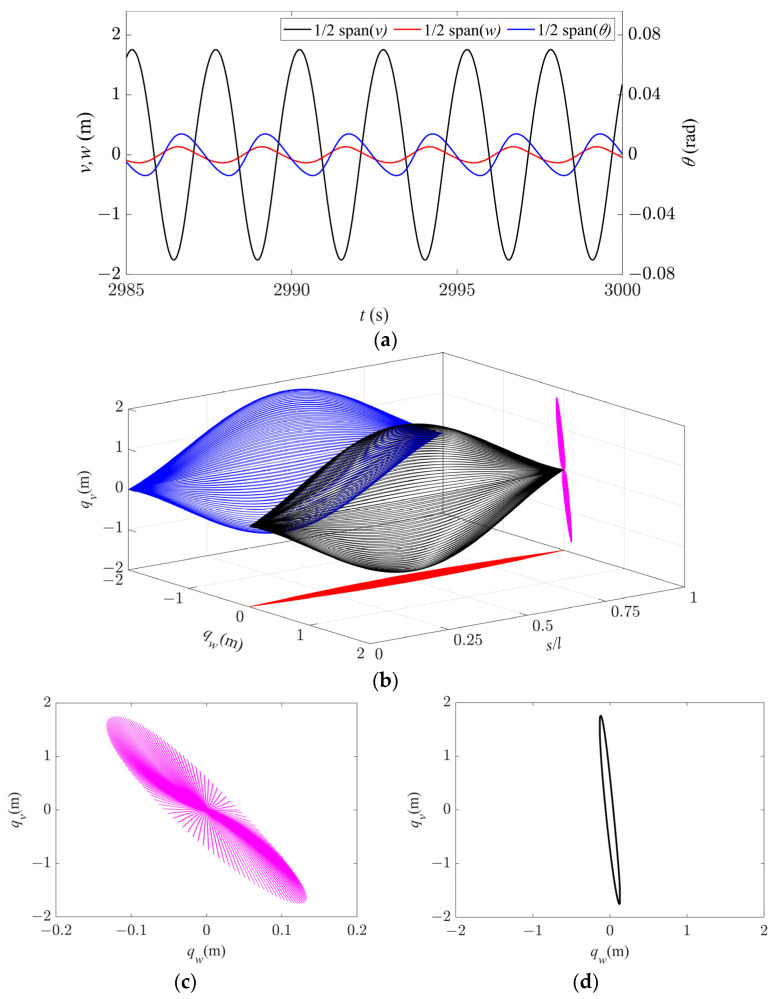
Galloping behavior in Region II (*d* = 4.3 m, *U* = 14 m/s). (**a**) Time history responses along the three directions at 1/2 span. (**b**) Spatial galloping trajectories along the three directions and their projections (*t* = 2997–3000 s). (**c**) Magnification of side view in (**b**). (**d**) Galloping orbit at 1/2 span.

**Figure 6 sensors-24-00784-f006:**
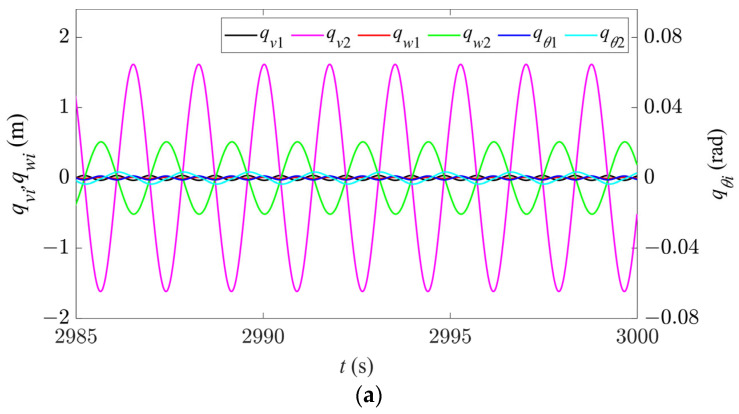
Time history and spectral responses of first- and second-order modal galloping along the three directions in Region III (*d* = 4.3 m, *U* = 19 m/s). (**a**) Time history responses of first- and second-order modal galloping along the three directions. (**b**) Spectral response of second-order modal galloping along the in-plane direction. (**c**) Spectral response of second-order modal galloping along the out-of-plane direction.

**Figure 7 sensors-24-00784-f007:**
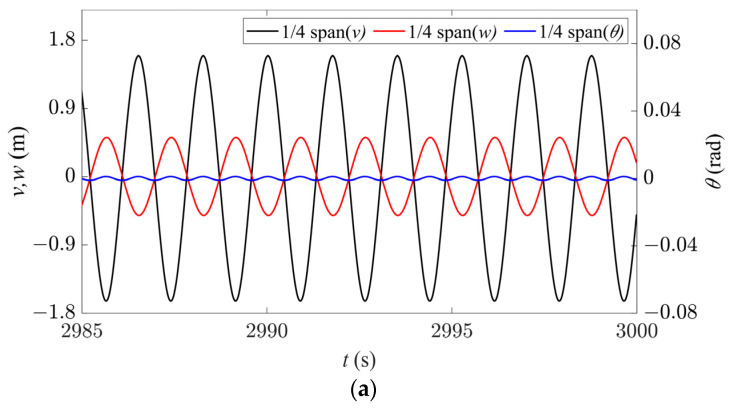
Galloping behavior in Region III (*d* = 4.3 m, *U* = 19 m/s). (**a**) Time history responses along the three directions at 1/4 span. (**b**) Spatial galloping trajectories along the three directions and their projections (*t* = 2996–2998 s). (**c**) Magnification of side view in (**b**). (**d**) Galloping orbit at 1/4 span.

**Figure 8 sensors-24-00784-f008:**
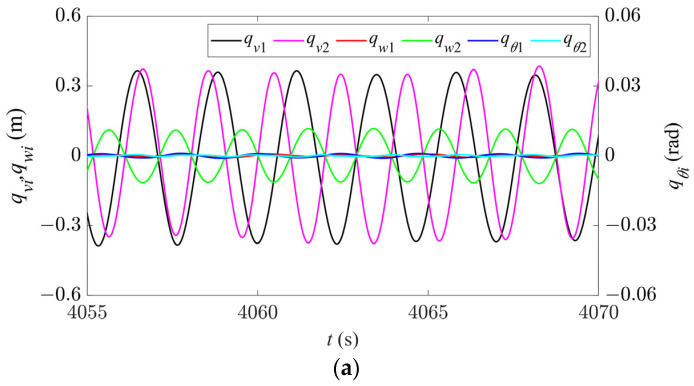
Time history and spectral responses of first- and second-order modal galloping along the three directions in Region IV (*d* = 4.6 m, *U* = 5.5 m/s). (**a**) Time history responses of first- and second-order modal galloping along the three directions. (**b**) Spectral response of first-order modal galloping along the in-plane direction. (**c**) Spectral response of second-order modal galloping along the in-plane direction. (**d**) Spectral response of second-order modal galloping along the out-of-plane direction.

**Figure 9 sensors-24-00784-f009:**
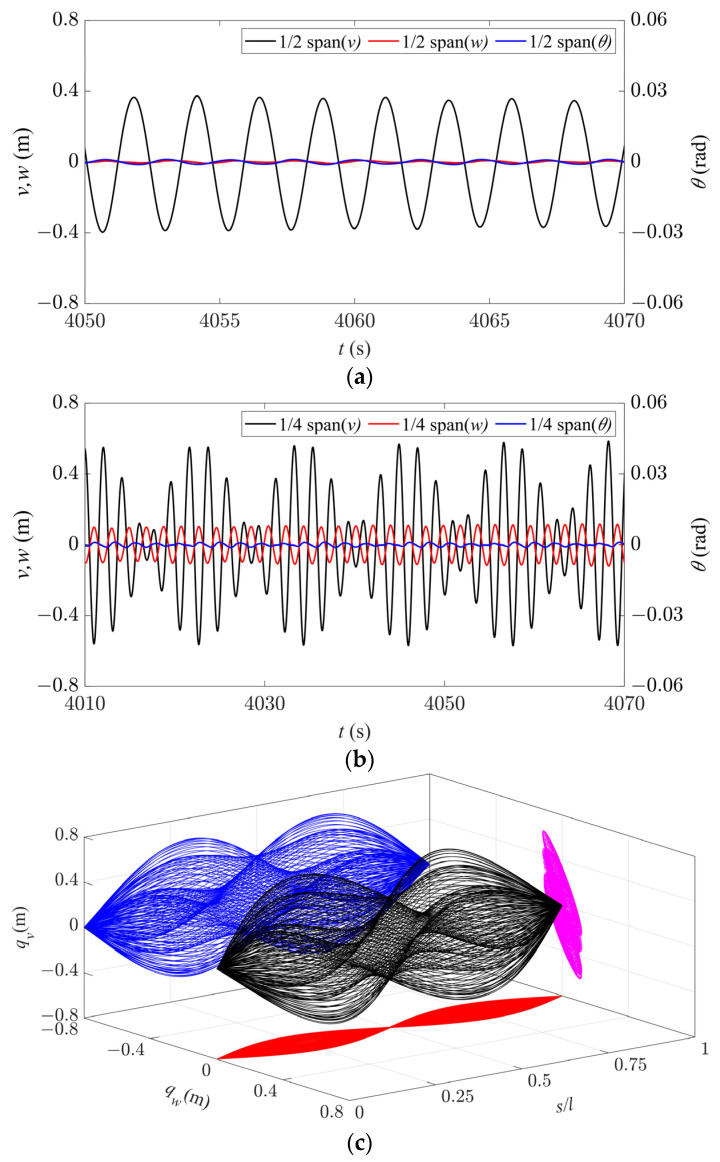
Galloping behavior in Region IV (*d* = 4.6 m, *U* = 5.5 m/s). (**a**) Time history responses along the three directions at 1/2 span. (**b**) Time history responses along the three directions at 1/4 span. (**c**) Spatial galloping trajectory along the three directions and the projections (*t* = 4052–4060 s). (**d**) Magnification of side view in (**c**). (**e**) Galloping orbit at 1/2 span. (**f**) Galloping orbit at 1/4 span. (**g**) Enlarged view of section A in (**f**).

**Figure 10 sensors-24-00784-f010:**
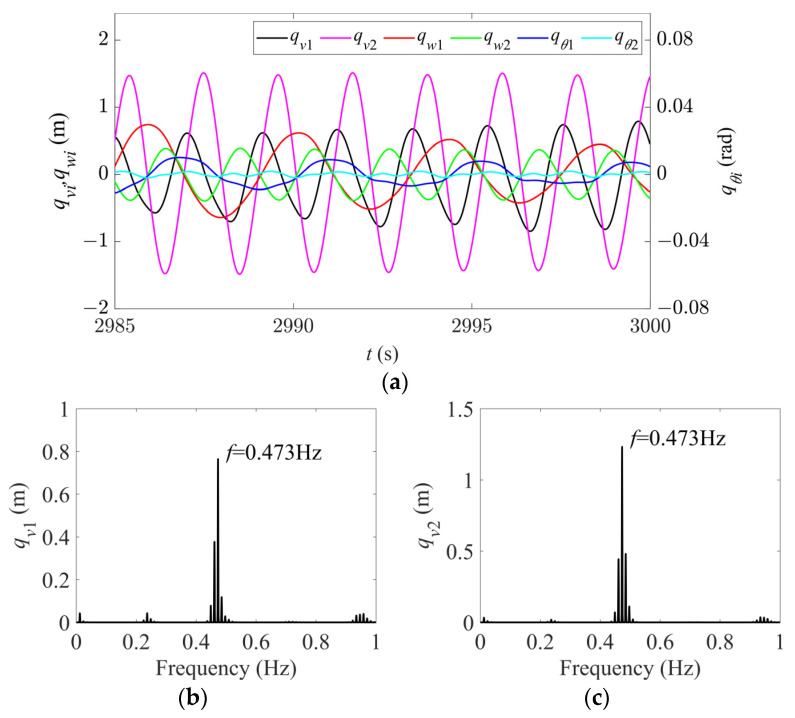
Time history and spectral responses of first- and second-order modal galloping along the three directions in region V (*d* = 6.8 m, *U* = 14 m/s). (**a**) Time history responses of first- and second-order modal galloping along the three directions. (**b**) Spectral response of first-order modal galloping along the in-plane direction. (**c**) Spectral response of second-order modal galloping along the in-plane direction. (**d**) Spectral response of first-order modal galloping along the out-of-plane direction. (**e**) Spectral response of second-order modal galloping along the out-of-plane direction. (**f**) Spectral response of first-order modal galloping along the torsional direction.

**Figure 11 sensors-24-00784-f011:**
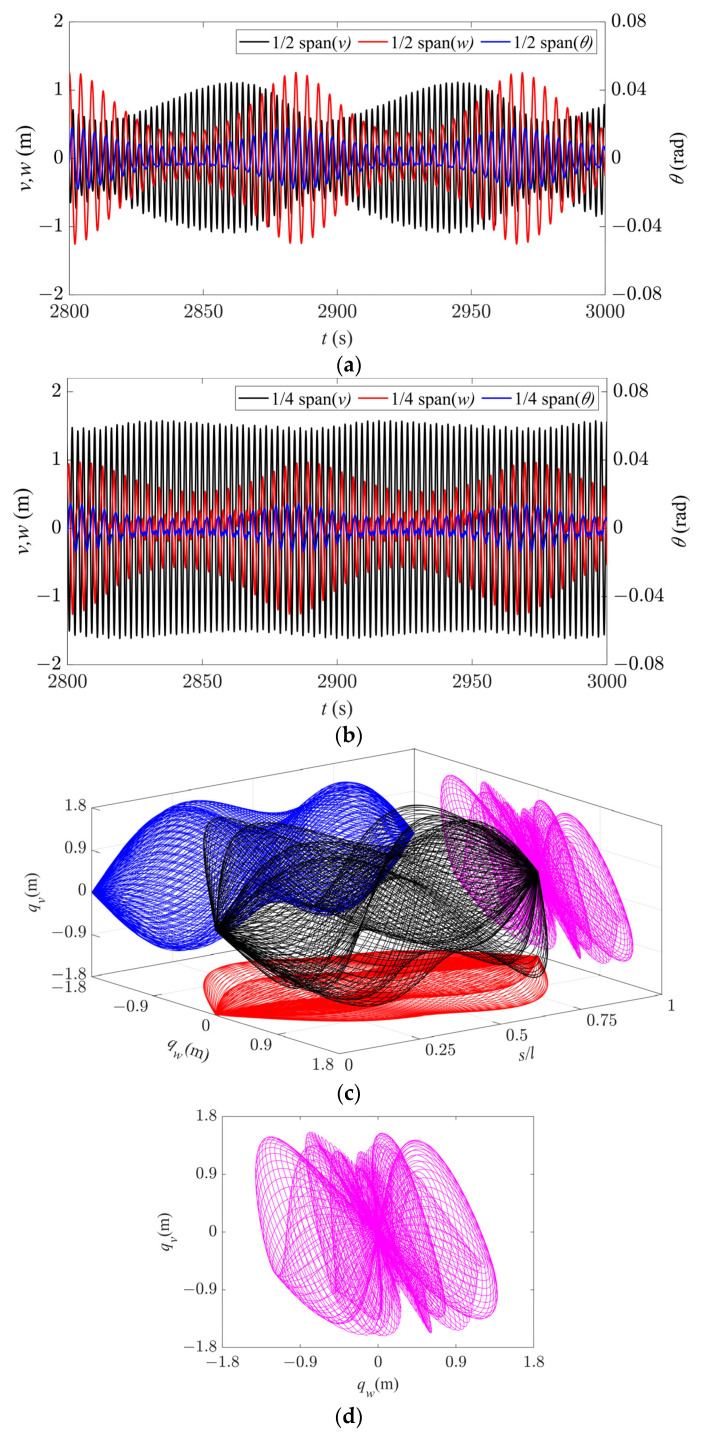
Galloping behavior in Region V (*d* = 6.8 m, *U* = 14 m/s). (**a**) Time history responses along the three directions at 1/2 span. (**b**) Time history responses along the three directions at 1/4 span. (**c**) Spatial galloping trajectories along the three directions and the projections (*t* = 2942–2970 s). (**d**) Magnification of side view in (**c**). (**e**) Galloping orbit at 1/2 span. (**f**) Galloping orbit at 1/4 span.

**Table 1 sensors-24-00784-t001:** Equivalent parameters of iced bundled conductors.

Parameter	Value
Span *l*	244 m
Mass per unit length *m*	6.92 kg/m
Bare conductor diameter *D*	28.6 mm
Tensile stiffness *EA*	1.105 × 10^8^ N
Torsional stiffness *GJ*	23,746 N·m^2^/rad
Moment of inertia *I*	0.70065 kg·m
Inflow density *ρ_a_*	1.29 kg/m^3^
Eccentricity *e_r_*	1.39 × 10^−4^ m
Initial wind attack angle *θ*_0_	24°
In-plane damping ratio *ξ_v_*	0.005
Out-of-plane damping ratio *ξ_w_*	0.005
Torsional damping ratio *ξ_ɵ_*	0.02

**Table 2 sensors-24-00784-t002:** First- and second-order natural frequencies along the three directions under different sags.

Direction	Natural Frequency					
	*d* = 4.3 m		*d* = 4.6 m		*d* = 6.8 m	
	1st	2nd	1st	2nd	1st	2nd
In-plane	0.421	0.534	0.433	0.516	0.516	0.424
Out-of-plane	0.267	0.534	0.258	0.516	0.212	0.424
Torsional	0.377	0.744	0.377	0.744	0.377	0.744

## Data Availability

Where data is unavailable due to privacy restrictions.

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
