# Peer review of "Spatial Galloping Behavior of Iced Conductors under Multimodal Coupling"

_sensors, 2024, doi:10.3390/s24030784_

Round 1
Reviewer 1 Report
Comments and Suggestions for Authors
In the references the many papers produced on conductors ice galloping and published by IEEE, Cigré and the Journal of Wind Engineering and Industrial Aerodynamics should be reported. For instance in paper [1] a FEM approach to reproduce galloping is presented
The authors should compare the results of the proposed approach to those obtained through models based on FEM.
In Figure 2.1 the authors define the type of ice considered. For this shape the aerodynamic coefficients used to compute the aerodynamic forces should be reported together with the approach used to define the aerodynamic forces themselves. No mention is made in the paper to the value of the Quasi Steady equilibrium position of the bundle under the wind load and ice weight. The wind load and the ice eccentricity produce a large rotation of the bundle and this plays a fundamental role in the galloping behaviour. The authors should comment on this matter.
[1] G.Diana, A.Manenti, S.Melzi "Energy method to compute the maximum amplitudes of oscillation of bundle conductors due to ice galloping" IEEE Transactions on Power Delivery - Print ISSN: 0885-8977 - Online ISSN: 1937-4208 - Digital Object Identifier: 10.1109/TPWRD.2020.3027157 - 2020
Reviewer 2 Report
Comments and Suggestions for Authors
The paper deals with a very important topic in the transmission line management. It is well written and the novelty introduced by authors is clearly expounded. In my opinion it deserves publication. However, I have these two issued should be considered before publishing the manuscript:
Comment 1)
If my understanding is correct, the mathematical model proposed by authors does not consider the presence of dampers installed along the line. How does the proposed formulation change by adding the presence of dumpers? This issue should be discussed in depth in the paper.
Comment 2)
Authors mentioned the importance of understanding and modelling the conductor galopping phenomenon, but it is also important to try to understand if such phenomenon is happening in real installed overhead lines or to try to predict such event. Could the proposed mathematical modelling approach be combined with real time measurements along the line to try to prevent the galloping event by means of a predictive algorithm based on the available measurement devices available on the market? Some comment about this issue would be interesting
Reviewer 3 Report
Comments and Suggestions for Authors
For the iced bundled crescent-shaped conductors, considering geometric and aerodynamic nonlinearity, authors construct the coupled vibration ordinary differential equations of the first two modes in three directions. By analyzing the characteristics of the solution of the nonlinear equation in the wind speed-sag parameter space, the critical conditions and spatial behavior characteristics of different galloping modes are obtained, and the mechanism of some galloping phenomena is explained, which has important theoretical significance for the formulation of effective anti-galloping measures.
Here are some suggestions that authors should consider to make this paper more interesting:
1. In the introduction (Page 2, Line 88-90), authors mention that the wind tunnel experiments have been conducted to verify the spatial nonlinear behavior of the galloping, but there is no corresponding content in this paper. I would suggest the authors to supplement the corresponding experiment results to enhance the persuasiveness of the article;
2. Page 3, Line 106, authors mention that the iced quad-bundled conductor is simplified as a single conductor for analysis. I would suggest the authors to provide the basis of the simplification or related references to support the rationality of the hypothesis;
3. Page 7, Line 218-220, authors mention that ‘In Region I … the vibration amplitudes of the first two modes in all the directions rapidly decay to zero …’, this is a typical result of damped vibration, while in the galloping behaviors of other regions, they are all constant amplitude vibration behaviors. I would suggest the authors to supplement the structural damping setting under various conditions, or point out the reasons for ignoring the damping effect to reduce the ambiguity of the article;
4. It can be seen from the results of galloping behaviors that authors should have set a reasonable time step to fully capture the response of the structure. How do authors balance the relationship between the solution time step and the fundamental frequency of the structure?
B. EDITORIAL COMMENTS
5. Table 3-1, the value of Tensile stiffness EA, number ‘8’ should be superscripted;
6. Table 3-1, the value of Eccentricity er, number ‘-4’ should be superscripted;
Page 6, line 207, number ' 3.1 ' should be changed to ‘3.2’.
Comments on the Quality of English LanguageMinor editing of English language required
Round 2
Reviewer 1 Report
Comments and Suggestions for Authors
The authors addressed my suggestions